# Alterations in the Milk Metabolome of Dairy Cows Supplemented with Different Levels of Calcium Propionate in Early Lactation

**DOI:** 10.3390/metabo12080699

**Published:** 2022-07-27

**Authors:** Fan Zhang, Yiguang Zhao, Hui Wang, Xuemei Nan, Yue Wang, Yuming Guo, Benhai Xiong

**Affiliations:** 1State Key Laboratory of Animal Nutrition, Institute of Animal Sciences, Chinese Academy of Agricultural Sciences, Beijing 100193, China; zhangfan19@139.com (F.Z.); zhaoyiguang@caas.cn (Y.Z.); wanghui_lunwen@163.com (H.W.); xuemeinan@126.com (X.N.); wangyue9313@163.com (Y.W.); 2State Key Laboratory of Animal Nutrition, College of Animal Science and Technology, China Agricultural University, Beijing 100193, China

**Keywords:** calcium propionate, Holstein dairy cow, negative energy balance, milk metabolites, early lactation

## Abstract

This study aimed to investigate the effects of dietary supplementation with different levels of calcium propionate on the lactation performance, blood energy metabolite parameters, and milk metabolites of dairy cows in early lactation. Thirty-two multiparous Holstein cows were randomly divided into 4 groups, which were orally drenched with 0, 200, 350, and 500 g/d calcium propionate per cow supplemented to a basal diet for 5 weeks from calving. The milk and blood of the dairy cows were sampled and measured every week. The milk samples from the last week were used for the metabolomic analysis via liquid chromatography–mass spectrometry (LC-MS). The results showed that the calcium propionate supplementation quadratically increased the dry matter intake, energy-corrected milk yield, and 4% fat-corrected milk yield; linearly reduced the milk protein and milk lactose concentrations; and quadratically decreased the somatic cell count in the milk. With the increase in calcium propionate, the serum glucose content showed a linear increase, while the serum insulin content showed a quadratic increase. The diets supplemented with calcium propionate quadratically decreased the β-hydroxybutyric acid and linearly decreased the non-esterified fatty acid content in the serum. The metabolomic analysis revealed that eighteen different metabolites were identified in the milk samples of the dairy cows supplemented with calcium propionate at 350 g/d, which decreased the abundance of genistein and uridine 5-monophosphate and increased the abundance of adenosine, uracil, protoporphyrin IX, and sphingomyelin (d 18:1/18:0) compared with the control group. The milk metabolic analysis indicated that the calcium propionate effectively improved the milk synthesis and alleviated the mobilization of adipose tissue and bone calcium. In summary, the calcium propionate could improve the lactation performance and energy status and promote the milk metabolic profile of dairy cows in early lactation. Calcium propionate (350 g/d) is a well-recommended supplement for dairy cows for alleviating negative energy balance and hypocalcemia in early lactation.

## 1. Introduction

In early lactation, dairy cows experience massive metabolic changes to support the rapid increases in milk production. Because of a transient decrease in feed intake, cows are challenged by negative energy balance (NEB) and substantial hypocalcemia around calving [1,2]. At parturition, the abrupt increase in lactose synthesis from 0 to approximately 1 kg/d in the mammary gland results in an increasing demand for glucose and a decrease in blood glucose [3]. The increasing glucose requirement stimulates fat mobilization to compensate for the energy deficit until approximately 40 to 80 d postpartum [4]. The energy deficit increases the ketone production and fat accumulation in the liver, leading to metabolic disorders of ketosis and fatty liver in early-lactation dairy cows. Excessive NEB will in turn induce a reduction in milk production. Cows with NEB are more susceptible to diseases such as mastitis, claw and leg diseases [5], acidosis, ketosis, and a decline in fertility [4,6]. The dairy cows with hypocalcemia transition to a positive calcium balance at approximately 6 to 8 weeks of early lactation [7]. Low-calcium diet or acidogenic diet strategies prepartum can improve the capacity for mobilizing calcium from the bones and can maintain calcium homeostasis in extracellular fluids [8]. However, these methods primarily depend on calcium mobilization from bone tissue, which causes the bone to be prone to spontaneous fracture, especially in older dairy cows [7]. The oral administration of large amounts of calcium salts has been proven to increase the blood calcium concentration by forcing calcium into the blood through passive diffusion [9,10].

In ruminants, propionate is the most predominant substrate for gluconeogenesis, a major pathway for maintaining an adequate glucose supply [11]. The increasing supplementation of propionate could alleviate NEB during early lactation. Calcium propionate, which can be hydrolyzed to propionic acid and Ca^2+^ in the rumen, is a good additive for alleviating NEB and the negative calcium balance of dairy cows in the perinatal period [9,12]. The supplementation of calcium propionate not only provides gluconeogenic precursors but also significantly increases the calcium intake by cows. The results of Martins et al.’s study [13] showed that Holstein cows fed calcium propionate had greater milk yields in early lactation. Liu et al. [10] also proved that increasing the supplementation of the diets to multiparous Holstein dairy cows with calcium propionate improved their energy status. In the previous study [14], we had investigated the effects of different feeding levels of calcium propionate on milk performance and serum metabolome in the last week after feeding for 5 weeks after calving, but the effects on the whole stage of early lactation in dairy cows still require further exploration. In particular, the application of metabolomics techniques to test the milk changes in calcium-propionate-supplemented dairy cows is still lacking.

Because the dairy cow diseases around parturition are related to lactation, involving a series of metabolic processes in the mammary gland, the milk metabolome (as a resulting product from lactation) may provide meaningful information on the altered metabolism in dairy animals, as well as the quality of milk under disease status [15]. Milk metabolomics based on ultra-high-performance quadrupole time-of-flight mass spectrometry (UHPLC-QTOF-MS) has been widely used to investigate changes in milk with different feeding regimes [16,17]. The non-invasively obtained milk samples can provide hints to the animal’s metabolic and health status [18,19]. Non-targeted metabolomics can detect metabolites with a relative molecular mass of less than 10,000 independent spectral features in biological samples [20], and can be used to explore the relationships between metabolites and physiological and pathological changes [21].

Therefore, the objective of this study was to investigate the effects of dietary supplementation with different doses of calcium propionate to dairy cows in the first 5 weeks after calving on lactation performance and blood metabolite parameters related to NEB at each week, and to monitor the changes in milk metabolites in the last week.

## 2. Results

### 2.1. Production Performance

The effects of the calcium propionate supplementation levels on DMI, body weight, milk yield, and milk composition values are shown in Table 1. The dry matter intake (DMI) (*p* = 0.02), energy-corrected milk (ECM) (*p* = 0.006), and 4% fat-corrected milk (FCM) (*p* = 0.008) values showed quadratic increases with the increasing supplementation of calcium propionate, and reached the highest values in the MCaP group. The milk protein (*p* = 0.007) and milk lactose (*p* = 0.003) values decreased linearly (*p* = 0.03) and the somatic cell count (SCC) (*p* = 0.03) showed a quadratic decrease. However, the calcium propionate supplementation did not affect the BW, milk fat concentration, or the ratio of milk fat to protein. The dynamic changes in DMI (A), milk yield (B), ECM (C), and 4% FCM (D) are also shown in Figure 1. The DMI, ECM, and 4% FCM values were all affected over time (*p* < 0.001), but treatment × time interaction effects were not observed. The cows in the MCaP group showed better DMI and milk production values during the experiment.

### 2.2. Serum Biochemical Parameters Related to NEB

The effects of dietary supplementation with different levels of calcium propionate on the serum parameters related to NEB are shown in Table 2. With increasing calcium propionate supplementation, the concentration of glucose increased linearly (*p* = 0.02), the insulin content showed a quadratic increasing trend (*p* = 0.08), and the values peaked in the MCaP group. The content of β-hydroxybutyric acid (BHBA) decreased quadratically (*p* = 0.003), and the non-esterified fatty acid (NEFA) concentration showed a linear decrease (*p* = 0.003) in increasingly calcium-propionate-supplemented cows. The dietary supplementation with calcium propionate did not affect the concentration of glucagon in the dairy cow serum. All of these parameters were affected over time (*p* < 0.001), but were not affected by the treatment × time interaction.

### 2.3. Milk Metabolomics Profiling

UHPLC-QTOF-MS was performed to detect the metabolites in the milk samples. The total ion chromatograms (TIC) of the quality control (QC) samples overlapped well and the adjacent peaks were well separated, which indicated that the liquid chromatography–mass spectrometry (LC-MS) methods in the experiment had high stability and repeatability and were suitable for sample identification (Appendix A). A multivariate data analysis was used to discover the changed milk metabolites and to explore the differences in milk produced between the CON and other calcium propionate treatment groups in early-lactation dairy cows.

A total of 83,979 peaks were detected in all samples, including 42,932 peaks in positive ion mode (ESI+) and 41,047 peaks in negative ion mode (ESI−) (Appendix A). After the metabolic data were filtered and normalized, 517 reliable metabolites were relatively quantified across all milk samples from the 4 groups (Appendix A). The principal component analysis (PCA) score plot showed that the CON treatment could be separated from the MCaP and HCaP groups. However, there was no clear distinction between the CON and LCaP groups (Appendix A). The reason may be related to the fact that all of the milk metabolites (including differential and non-differential ones) were included in the analysis. The supervised orthogonal projections to latent structures discriminate analysis (OPLS-DA) (Figure 2) helped visualize the group separation process and to identify the differential metabolites of milk samples from different groups. In the results for milk production, we found that the calcium propionate quadratically increased the milk yield. Therefore, a univariate analysis of the metabolomic profile was performed, including not only between the CON group and the other calcium propionate treatment groups (LCaP, MCaP, and HCaP groups) but also between the MCaP and HCaP groups. As shown in Figure 2, the supervised OPLS-DA method presents clear distinctions among the four groups.

### 2.4. Differences in Milk Metabolites

The differential milk metabolites were identified with a cutoff value of variable importance in the projection (VIP) > 1 and *p* < 0.05. Based on the results of the LC-MS metabolomics analysis, 69 metabolites were obtained from the comparison of LCaP compared to CON, MCaP compared to CON, HCaP compared to CON, and HCaP compared to MCaP (Appendix A and Table 3). In total, there were 8 differential metabolites between LCaP and CON, of which 6 differential metabolites were downregulated and 2 differential metabolites were upregulated in the LCaP compared to the CON group (Appendix A). There were 18 differential metabolites between MCaP and CON, of which 11 differential metabolites were downregulated and 7 differential metabolites were upregulated in the milk samples of the MCaP compared to the CON group (Appendix A). Compared with the CON group, 22 differential metabolites were found, of which 18 differential metabolites were downregulated and 4 differential metabolites were upregulated (Appendix A). In the comparison of HCaP compared to MCaP, 11 differential metabolites were downregulated and 10 differential metabolites were upregulated (Appendix A).

Table 3 and Figure 3 show the differential metabolites in the milk samples between the treatment groups. Compared with the CON group, the main differential metabolites of 4-pyridoxic acid and N-acetyl-D-glucosamine 6-phosphate were downregulated, but the metabolites of traumatic acid and Trp-Glu-Arg were upregulated in the LCaP group. In the MCaP group, the main differential metabolites of genistein and alpha-ketoisovaleric acid were decreased, while the main differential metabolites of adenosine, uracil, protoporphyrin IX, and sphingomyelin (d 18:1/18:0) were increased compared with the CON group. The main differential metabolites of Pro-Val, D-mannitol, (2-(2-methoxyethoxy) ethoxy)-acetic acid, DL-lactate, and D-glucuronate were downregulated, but the main differential metabolite of protoporphyrin IX was upregulated in the HCaP group when compared with the CON group. The main differential metabolites of 1-octadecyl-2-acetyl-sn-glycero-3-phosphocholine, 1-hexadecyl-sn-glycero-3-phosphocholine, 1-hexadecanoyl-sn-glycero-3-phosphoethanolamine, and adenosine were downregulated, but the main differential metabolites of N-(4-hydroxyphenyl) propenamide, Leu-Gly, and 1,5-diaminonaphthalene were upregulated in the HCaP group when compared with the MCaP group. As shown in Figure 3 and Figure 4, 1 mutual metabolite (N-carboxyethyl-gamma-aminobutyric acid) was identified between the differential metabolites in LCaP compared to CON and MCaP compared to CON. Five mutual metabolites (2-methylglutaric acid, uridine 5-monophosphate, o-amino-phenol, protoporphyrin IX, and D-(+)-galactose) were identified between the differential metabolites in MCaP compared to CON and HCaP compared to CON. The detailed information about the differential metabolites is shown in Appendix A.

### 2.5. Metabolic Pathways Enrichment of Differential Metabolites

The metabolic pathways were analyzed based on the identified differential metabolites and their expression levels between the different treatment groups (Figure 5). The pathway of vitamin B6 metabolism (*p* = 0.079, impact value = 0.036) was enriched from the differential metabolites between the LCaP and CON groups. The pathways of pantothenate and CoA biosynthesis (*p* = 0.009, impact value = 0.071), beta-alanine metabolism (*p* = 0.011, impact value = 0.063), morphine addiction (*p* = 0.041, impact value = 0.125), and the cGMP-PKG signaling pathway (*p* = 0.051, impact value = 0.100) were mainly enriched from the differential metabolites between MCaP and CON groups. The differential metabolites between the HCaP and CON groups were mainly enriched in the AMPK signaling pathway (*p* = 0.011, impact value = 0.091) and pyrimidine metabolism pathway (*p* = 0.085, impact value = 0.031). The differential metabolites between the HCaP and MCaP groups were mainly enriched in the cGMP-PKG signaling pathway (*p* = 0.001, impact value = 0.200).

## 3. Discussion

### 3.1. Animal Performance

Dairy cows in early lactation often continue in an NEB state because of the increased energy requirements for milk production compared to the relatively low energy intake. High-producing dairy cows are at high risk for severe NEB [23]. It has previously been reported that feeding with calcium propionate [10] or gluconeogenic precursors [24] does not affect DMI. In the current study, however, the DMI increased quadratically with the increasing calcium propionate supplementation. This may be related to the observations that calcium propionate supplementation improved milk production and alleviated the NEB status of the dairy cows. With the increasing supplementation of calcium propionate to 500 g/d per cow, the DMI of the dairy cows in the HCaP group was decreased compared with that in the MCaP group, which may be related to the hypophagic effects of propionate [25] and the negative impact of the higher dietary calcium intake on the feed intake [26]. When the dairy cows were fed propionate over the required amount, it inhibited the feed intake. Feeding with excessive dietary calcium by more than 1% has been associated with reduced DMI and lower performance [7]. In the current study, the BW values of the 4 groups of cows showed no significant differences with calcium propionate supplementation, which was consistent with the previous studies that calcium propionate supplementation [10] and rumen-protected glucose [27] intake did not affect the BW values of dairy cows during early lactation.

The milk yield of dairy cows is mainly related to the gluconeogenesis precursor of propionate. In the current study, cows treated with calcium propionate had a higher milk yield than the other cows. The cows in the calcium-propionate-treated groups received extra propionate, which could result in more glucose produced in the liver [28]. The increased glucose production alleviated the NEB of the dairy cows and improved their milk production. This finding fitted well with the reports by Lomander et al. [29] and Melendez et al. [28], in which oral the administration of gluconeogenic precursors improved the milk yield and maintained a moderate energy status during the transition period. With the increased feeding levels of calcium propionate from 350 to 500 g/d, the milk yield was decreased, which might be associated with the high calcium propionate feeding level inhibiting the appetite of the dairy cows. Feeding with calcium propionate reduced the milk protein and milk lactose concentrations in the dairy cows, which may be related to the increased milk production. In the present study, the increase in milk production and the decrease in SCC might also be partially due to the improved immune function of the gluconeogenic precursor and the calcium source of the calcium propionate. Supplementing calcium propionate to the dairy cows in the transition period decreased the NEB [2] and provided a calcium source [30], which was beneficial in improving their immunity and avoiding hypocalcemia [9]. Subsequently, it was beneficial for increasing their milk yield and reducing milk SCC. From the perspective of cow-performance-related indices, 350 g/d calcium propionate was the optimum dose for dairy cows in early lactation.

### 3.2. Serum Biochemical Parameters Related to NEB

During the initial lactation period, the dairy cows experience a period of NEB because of an insufficient dietary intake [31]. To compensate for an energy deficit, cows mobilize body fat and release NEFA into circulation [32]. Serum glucose, NEFA, and BHBA are important indicators of energy metabolism in dairy cows, and researchers have reported that plasma NEFA and BHBA correlate negatively with energy balance in dairy cows [33].

Dietary supplementation with calcium propionate could serve as a gluconeogenic precursor when the cows are in NEB conditions. As an important substitute for hepatic gluconeogenesis, the contribution range of calcium propionate is approximately 60% to 75% [34]. Supplementation with calcium propionate improved the serum glucose and insulin concentrations, demonstrating the glucogenic capacity of this additive. Kennedy et al. [35] reported that a propionate supply given to dairy cows postpartum not only upregulated gluconeogenesis and the pool of acetyl-CoA but also stimulated acetyl-CoA from fat decomposition to be effectively oxidized, which avoided the production of ketone bodies [36]. The results of the study by Goff et al. [9] showed that calcium propionate supplementation at calving and 12 h after calving tended to reduce plasma NEFA and BHBA levels. It was observed that feeding 100 to 300 g/d calcium propionate [10] increased the plasma glucose and insulin concentrations in early-lactation cows. In the current study, the MCaP treatment improved the serum glucose and reduced the BHBA and NEFA levels, which was consistent with the previous research showing that calcium propionate supplementation was beneficial for alleviating NEB and decreasing fat mobilization in postpartum dairy cows. 

### 3.3. Differences in Milk Metabolites and the Metabolic Pathway Enrichment Analysis

The characterization of the milk metabolome is a promising approach for establishing its overall quality and authenticity [37]. Xu et al. [38] reported that the energy balance of dairy cows in early lactation was highly correlated with several milk metabolites, including glycine, choline, and carnitine. Milk metabolomics could be a useful tool for the continuous monitoring of herd health [18]. Thus, in the present study, we investigated the milk metabolites to better understand the function and metabolic status of calcium propionate supplementation in the early-lactation dairy cows.

Our metabolome data revealed that calcium propionate altered the concentrations of metabolites in the milk, indicating that the milk metabolism might be related to the nutritional function of calcium propionate supplementation in early-lactation dairy cows. As the major degradation product of vitamin B6, the decrease in the 4-pyridoxic acid level at least in part reflected impaired kidney function and inflammation, which decreased the vitamin B6 turnover. The vitamin B6 nutrition metabolism was closely related to the alkaline phosphatase (ALP) level [39]. Therefore, the decreased 4-pyridoxic acid level may be related to the decreased ALP level in the LCaP group, which was discovered in our previous study [14]. N-Acetyl-D-glucosamine 6-phosphate is an allosteric modulator of deaminase, which plays a key role in the control of amino–sugar utilization [40]. The decrease in this metabolite in the LCaP group indicated that the feeding of calcium propionate inhibited the gluconeogenesis from the amino acids. The metabolite of N-carboxyethyl-gamma-aminobutyric acids, which was reduced in LCaP and MCaP groups compared with the CON group, was only preliminarily observed to have growth-promoting activity in some specialized mammalian cells [41]. The function of the metabolite in the current study was not obvious. 

Genistein has a crucial role in regulating glucose and lipid metabolism with the properties of antioxidants, downregulating fasting blood glucose and decreasing adipose deposition levels [42]. The decreased concentration of genistein in milk may be related to the function of calcium propionate in improving the serum glucose concentration and decreasing the serum antioxidants. Adenosine could promote the osteogenic ability and exert a negative effect on the osteoclastogenesis of osteoclast precursors [43]. The adenosine concentration was increased in the MCaP group when compared with the CON group, while it was decreased in the HCaP group when compared with the MCaP group. The increased concentration of adenosine in the milk of the MCaP group revealed that the supplementation of calcium propionate alleviated the mobilization of bone calcium. The cGMP-PKG signaling pathway related to the metabolites of adenosine and guanosine 5-monophosphate has also been reported to play important roles in modulating neuroinflammation, oxidative stress, and apoptosis [44] and promoting osteogenesis [45]. As pyrimidine bases, the change in the uracil content in milk is regarded to be related to milk synthesis [46] and SCC [47]. The increased concentration of uracil in the MCaP group may be due to the increased milk production. As one kind of choline pool, the increase in the sphingomyelin concentration could enhance very low density lipoprotein assembly to reduce liver triglyceride deposition in the lactating cows [48]. In this study, the observed higher levels of sphingomyelin in the MCaP group indicated that calcium propionate effectively inhibited the fatty liver at this feeding level. Alpha-ketoisovaleric acid is a keto-acid that arises from the incomplete breakdown of branched-chain amino acids, which is induced by the symptoms of poor feeding, loss of appetite, weak muscle tone (hypotonia), and lack of energy. Abnormally high levels of alpha-ketoisovaleric acid lead to general metabolic acidosis. The decreased concentration of alpha-ketoisovaleric acid indicated the function of calcium propionate in improving the metabolic state. Furthermore, the large amounts of protoporphyrin IX are hepatotoxic, damaging both the hepatocytes and cholangiocytes [49], which corresponds with our previous study that high calcium propionate supplementation impaired the liver function and increased the oxidative stress [14]. The AMPK signaling pathway plays a critical role in energy homeostasis by promoting fatty acid oxidation and autophagy. The metabolic pathway enrichment analysis revealed that the AMPK signaling pathway was inactivated in the HCaP group when compared with the CON group, which means that the calcium propionate effectively inhibited the rate of lipolysis. Therefore, we estimated that the supplementation of calcium propionate to the dairy cows at that optimum level (350 g/d) may be beneficial in improving the metabolic status from the results of the milk metabolite tests. However, the specific mechanisms underlying the calcium propionate regulation of milk metabolism in early lactation require further investigation.

## 4. Materials and Methods

This study was carried out according to the recommendations and principles of the Chinese Academy of Agricultural Sciences Animal Care and Use Committee (Beijing, China). The experimental protocols and methods involving animals were approved by the Animal Ethics Committee of the Chinese Academy of Agricultural Sciences (permit number: IAS2020-93). The experiments were performed in a commercial dairy farm (Beijing, China) from September to December 2020. The procedures used for sample collection and the chemical analysis were similar to the previous study by Zhang et al. [14].

### 4.1. Animals and Experimental Design

In a randomized complete block design, 32 multiparous Holstein cows with similar parity (3.29 ± 0.18), body weight (779 ± 12.5 kg), and previous lactation milk yield (41.61 ± 1.61 kg/d) values were randomly assigned to 4 treatments with 8 replicates; each group was assigned based on parity, body weight, milk yield, and anticipated calving date. The treatments, including control (CON), low calcium propionate (LCaP), medium calcium propionate (MCaP), and high calcium propionate (HCaP), were provided at the same total mixed ration (TMR) with different doses of calcium propionate (0, 200, 350, and 500 g/d per cow). The food-grade calcium propionate (Jiangsu Runpu Food Technology Co., Ltd., Lianyungang, Jiangsu, China) was given as an oral drench 3 times a day in equal amounts to the Holstein cows after milking. All cows in the experiment had free access to water and feed.

The TMR was formulated and prepared according to the National Research Council [7] requirements. The TMR was offered 3 times a day (7:00, 14:30, and 18:00). The ingredients and chemical compositions of the TMR are shown in Table 4. The orts were collected and weighed before the morning feeding. The TMR was offered ad libitum with 5% to 10% orts every day. The trial lasted for 35 days, and the calcium propionate supplementation was administered stepwise to reach the desired amount for each group in the first 3 days. Three cows were eliminated for metritis (one cow in the MCaP group) and left displaced abomasum (two cows in the CON group) in the experiment, which were not included in the data analysis. The dairy cows were weighed every week before the morning feeding.

### 4.2. Feed Samples and Chemical Analysis

Every day, the amounts of feed offered and refused were measured to calculate the DMI. The feed samples were collected weekly from 5 different feed bunk locations at feeding time for the analysis of the TMR chemical compositions. All of the samples were analyzed for dry matter (DM) (method 934.01) [50], crude protein (CP) (method 954.01) [50], ether extract (EE) (method 920.39) [50], calcium (Ca) (method 968.08) [50], phosphorous (P) (method 946.06) [50], and acid detergent fiber (ADF) (method 973.18) [50] contents. The neutral detergent fiber (NDF) was analyzed according to the method used by Van Soest et al. [51] with heat-stable α-amylase and expressed inclusive of residual ash. The net energy of lactation (NEL) was calculated according to NRC [7].

### 4.3. Milk Sample Collection and Laboratory Analysis

The cows were milked 3 times a day (6:00, 14:00, and 22:00), and the milk yield (kg/d) of each cow was recorded automatically daily using the AfiMilk MPC milk meter (2 × 14 fishbones milking machine, AfiMilk Ltd., Kibbutz Afikim, Israel). The milk samples were collected at days 7, 14, 21, 28, and 35 postpartum. The milk samples of each cow were collected at 3 consecutive milking periods (6:00, 14:00, and 22:00) on the sampling day and mixed at a ratio of 4:3:3 as a weekly sample. These milk samples were stored in plastic containers (50 mL) with a preservative (2-bromo-2-nitropropan-1,3-diol) at 4 °C until the determination of the SCC and milk composition. The milk samples were analyzed using the CombiFoss 4000 (Foss Electric A/S, Hillerød, Denmark) for fat, protein, lactose, and SCC contents within 24 h. The ECM yield (kg/d) was calculated as ECM = 0.327 × milk yield (kg/d) + 12.95 × fat yield (kg/d) + 7.20 × protein yield (kg/d) [22] and the 4% FCM yield (kg/d) was calculated as 4% FCM = 0.4 × milk yield (kg/d) + 15 × milk fat (kg/d) [7]. 

### 4.4. Blood Sample Collection and Analysis

The blood samples from each cow were taken by puncturing the coccygeal vein using a 10 mL gel vacuum within 1 h of the morning feeding on days 7, 14, 21, 28, and 35 postpartum. The samples were centrifuged at 3000× *g* for 15 min at 4 °C to collect the serum, which was stored at −20 °C until the subsequent analysis. The serum glucose was measured using the glucose oxidase enzymatic colorimetric method (Shanghai Kehua Bio-Engineering Co., Ltd., Shanghai, China) and the results were determined at 500 nm using a microplate reader (DNM-9602, Perlong Medical Co., Beijing, China). The intra- and inter-assay coefficients of variation (CVs) for glucose were below 4% and 5%, respectively. The concentrations of insulin, glucagon, and NEFA were determined using commercial ELISA kits (Nanjing Jiancheng Bioengineering Institute, Nanjing, China) at a wavelength of 450 nm. The intra- and inter-assay CVs for insulin, glucagon, and non-esterified fatty acids (NEFA) were all <10% and <15%, respectively. The BHBA concentration was assayed using a kinetic-enzymatic method (Shanghai Kehua Bio-Engineering Co., Ltd., Shanghai, China) at a wavelength of 340 nm. The intra- and inter-assay CVs for BHBA were below 6% and 8%, respectively.

### 4.5. LC-MS Metabolomics Processing

#### 4.5.1. Metabolomics Processing and Multivariate Statistical Analysis

In this study, the milk metabolites were analyzed via LC-MS. The detailed procedure for the LC-MS analysis was performed in our previous study [14]. In brief, six cows’ milk samples from the last week (d 35 postpartum) from each group were randomly assigned for the metabolomics analysis. A total of 24 milk samples were thawed and mixed thoroughly before LC-MS/MS analyses. Approximately 100 μL milk samples were transferred to a 2 mL Eppendorf tube and resuspended with 1000 μL ice-cold extraction solvent (acetonitrile: methanol: water, 2:2:1) by vortexing for 30 s. Then, the mixture samples were homogenized for 10 min and incubated at −20 °C for 1 h. After centrifugation (13,000 rpm, 4 °C, 15 min), 350 μL of the supernatant was transferred to a 1.5 mL Eppendorf tube and then dried in a vacuum concentrator. The mixture was redissolved in extract solvent (acetonitrile-water, 1:1), vortexed for 30 s, placed under ultrasound for 10 min, then centrifuged at 13,000 rpm at 4 °C for 15 min. Finally, 50 µL of supernatant was transferred to LC vials for subsequent UHPLC-QTOF-MS analysis.

The UHPLC-QTOF-MS metabolomics analysis was performed using an Exion LC™ AC system (SCIEX, Concord, NH, USA) coupled with a Triple TOF 5600+ quadrupole-time-of-flight mass spectrometer (SCIEX, Concord, NH, USA). A Waters UPLC column (1.7 µm, 2.1 × 100 mm, Waters, Milford, MA, USA) was used for the chromatographic separation and the column temperature was maintained at 40 °C. The mobile phase consisted of 25 mmol/L ammonium acetate and 25 mmol/L ammonium hydroxide in water (A), and 100% acetonitrile (B); the elution gradient was as follows: 0–0.5 min, 95% B; 0.5–7 min, 95–65% B; 7–8 min, 65–40% B; 8–9 min, 40% B; 9–9.1 min, 40–95% B; and 9.1–12 min, 95% B. The flow rate was set to 0.5 mL/min and the injection volume was 2 µL. The metabolite identification was performed using an electrospray ionization (ESI) source in positive and negative ion modes. The QTOF-MS system conditions for the positive ion mode (ESI+) and negative ion mode (ESI−) were at 5500 V and −4500 V, respectively. The ESI source conditions were as follows: ion gas temperature at 650 °C, ion gas pressure 1 at 60 psi, ion source gas 2 at 60 psi, curtain gas at 30 psi, and decluttering potential of 60 V. In each data collection cycle, molecular ions with the intensity levels greater than 100 were selected to collect the corresponding secondary mass spectrometry data. The QTOF-MS scanning range was 60–1200 mass-to-charge ratio (*m*/*z*) at 0.15 s/spectra using a collision energy of 10 eV. The MS/MS data were recorded with a *m*/*z* range of 25–1200 at 0.03 s/spectra and a collision energy of 30 eV.

#### 4.5.2. Metabolomics Data Analysis

MS raw data files were converted into mzXML format using the ProteoWizard software. Then, the R package of XCMS (V.3.2) was used to obtain the data retention time (RT), *m*/*z* values, and peak intensity for metabolite identification. A metabolite was detected if the feature was <20% of experimental samples or <50% of QC samples (prepared by mixing sample extracts). The missing values for the raw data were filled up to half of the minimum value. Finally, features with a relative standard deviation of QC > 30% were removed from the subsequent analysis. The internal standard normalization method was used in the data analysis. After XCMS data processing, the R package CAMERA was used for peak annotation. The online HMDB (http://www.hmdb.ca (accessed on 20 March 2021)), PubChem, Chemical Abstracts Service (CAS), Kyoto Encyclopedia of Genes and Genomes (KEGG), and the Chemical Entities of Biological Interest (ChEBI) databases were used for metabolite identification and validated by aligning the molecular mass data (*m*/*z*) [52]. The SIMCA 14.1 software package was used for the multiple statistical analysis consisting of a PCA and OPLS-DA. The different metabolites between the different calcium propionate treatments were identified based on pairwise comparisons. The first principal component of the VIP was obtained to refine the analysis. The metabolites with VIP > 1 and *p* < 0.05 (assessed by student *t*-test) were considered significantly changed metabolites. MetaboAnalyst 4.0 (http://www.metaboanalyst.ca/ (accessed on 22 March 2021)) was used to search for the pathways of different metabolites in each comparison according to the KEGG database (http://www.genome.jp/kegg/ (accessed on 22 March 2021)).

### 4.6. Statistical Analysis

The statistical power analysis for the sample size was above 0.8, with a significance level of 0.05 in G*Power software (version 3.1.9.6, https://g-power.apponic.com (accessed on 10 November 2020)) in this study. The data were tested for normality using the UNIVARIATE procedure of SAS 9.4 (SAS Institute Inc., Cary, NC, USA). The DMI and milk production data were averaged to weekly means for the statistical analysis. The MIXED procedure from SAS 9.4 was used to analyze the data in a randomized block design, which included repeated measures (DMI, BW, milk production, milk composition, and serum metabolites). The repeated measures model contained the fixed effects of treatment, time, treatment × time, and the random effect of the block. The statistical model followed the formula: Y_ijk_ = u + α_i_ + β_j_ + γ_k_ + α_i_ × β_j_ + ε_ijk_,
where Y_ijk_ was the dependent variable, u was the overall mean, α_i_ was the fixed effect of treatment, β_j_ was the fixed effect of time, γ_k_ was the random effect of the block, α_i_ × β_j_ was the interaction of the treatment and time, and ε_ijk_ was the random error. The results were presented as the least square means and standard errors of the mean. The linear and quadratic effects of the dietary supplementation with calcium propionate were determined via the orthogonal polynomial contrasts. The IML procedure from SAS was used to generate contrast coefficients adjusted for the unequal spacing of the calcium propionate supplementation in the diet. A significant difference was considered at *p* ≤ 0.05, and a tendency was accepted at 0.05 < *p* ≤ 0.10.

## 5. Conclusions

The present study showed that dietary supplementation with calcium propionate to dairy cows in early lactation for 5 weeks could effectively increase the DMI and milk production and alleviate NEB. Dietary supplementation with 350 g/d to dairy cows in early lactation was the optimum feeding level. In addition, eighteen differential metabolites were identified at this feeding level via the milk metabolic analysis compared with the CON group. The altered metabolic processes explained some of the mechanisms of calcium propionate in effectively improving the milk synthesis and alleviating the mobilization of adipose tissue and bone calcium. The results provided the basis for calcium propionate application in dairy cows for alleviating negative nutrient balance in early lactation.

## Figures and Tables

**Figure 1 metabolites-12-00699-f001:**
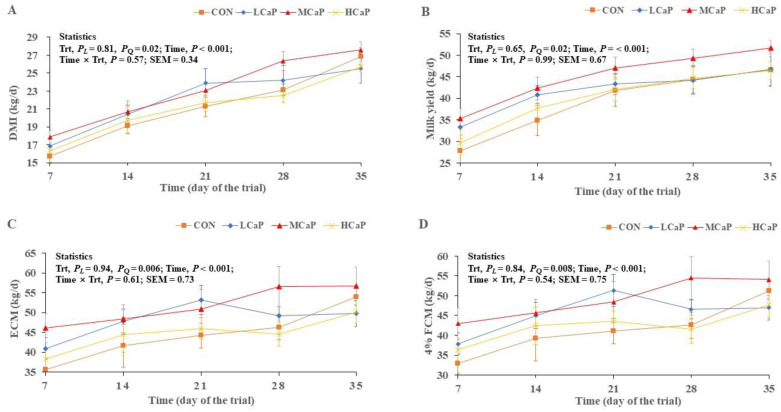
Dynamic changes in DMI (**A**), milk yield (**B**), ECM (**C**), and 4% FCM (**D**) after supplementing dairy cows with calcium propionate in early lactation. CON: control group, without calcium propionate addition; LCaP: low calcium propionate, the calcium propionate addition level was 200 g/d per cow; MCaP: medium calcium propionate, the calcium propionate addition level was 350 g/d per cow; HCaP: high calcium propionate, the calcium propionate addition level was 500 g/d per cow. Trt = treatment; PL = linear effects of the treatment; PQ = quadratic effects of the treatment; Time = time effect; Time × Trt = the interaction effect between time and treatment; SEM = standard error of the mean.

**Figure 2 metabolites-12-00699-f002:**
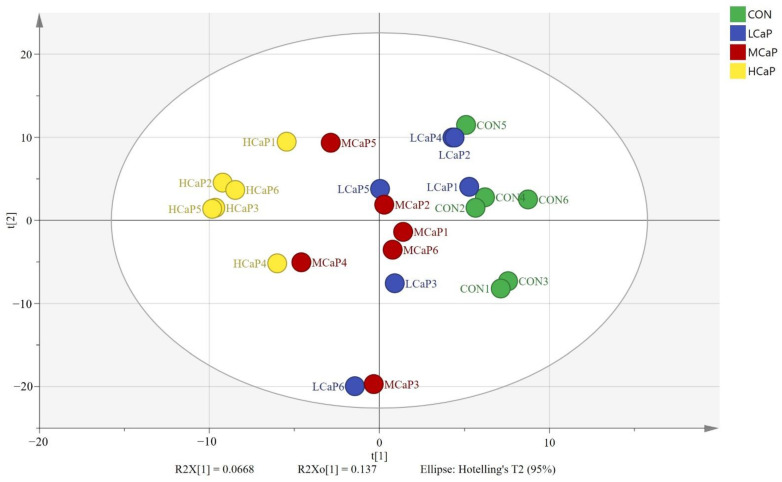
Orthogonal projections to latent structures discriminant analysis (OPLS-DA) score plot for the milk samples collected from different feeding levels of calcium propionate in early-lactation dairy cows. CON: control group, without calcium propionate addition; LCaP: low calcium propionate, the calcium propionate addition level was 200 g/d per cow; MCaP: medium calcium propionate, the calcium propionate addition level was 350 g/d per cow; HCaP: high calcium propionate, the calcium propionate addition level was 500 g/d per cow.

**Figure 3 metabolites-12-00699-f003:**
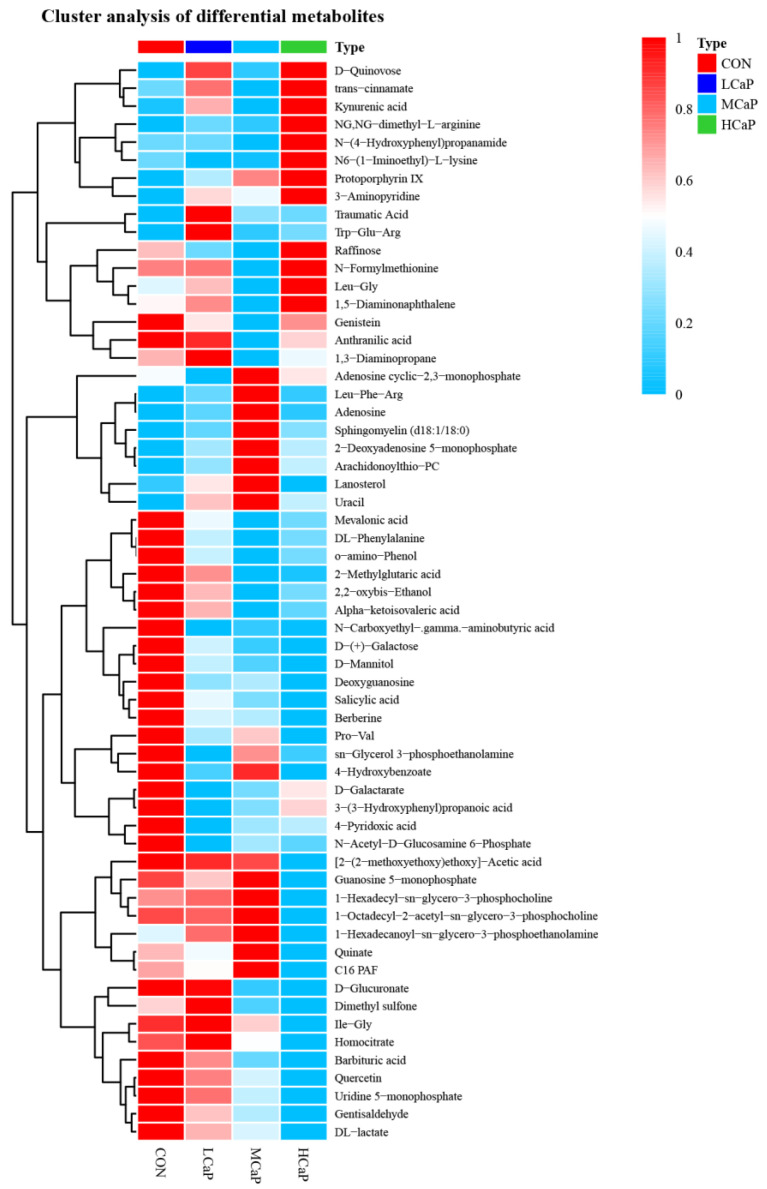
Hierarchical clustering analysis for different metabolites of the milk samples collected from different feeding levels of calcium propionate in early-lactation dairy cows. CON: control group, without calcium propionate addition; LCaP: low calcium propionate, the calcium propionate addition level was 200 g/d per cow; MCaP: medium calcium propionate, the calcium propionate addition level was 350 g/d per cow; HCaP: high calcium propionate, the calcium propionate addition level was 500 g/d per cow.

**Figure 4 metabolites-12-00699-f004:**
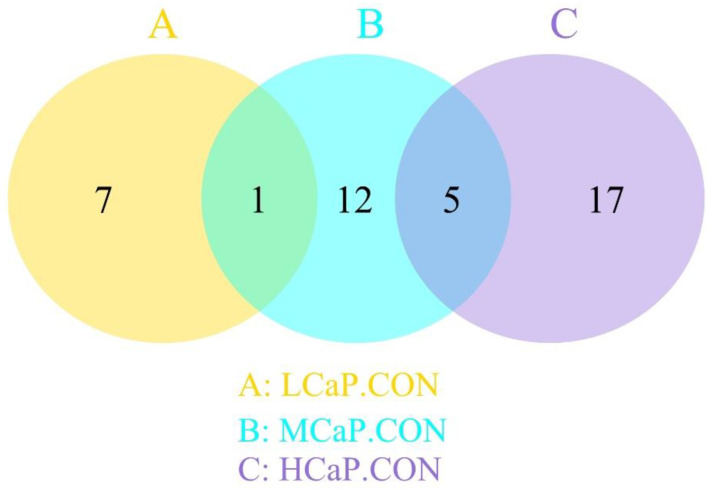
Venn diagrams of differential metabolites for pairwise comparison between the CON group and the LCaP, MCaP, and HCaP groups. CON: control group, without calcium propionate addition; LCaP: low calcium propionate, the calcium propionate addition level was 200 g/d per cow; MCaP: medium calcium propionate, the calcium propionate addition level was 350 g/d per cow; HCaP: high calcium propionate, the calcium propionate addition level was 500 g/d per cow.

**Figure 5 metabolites-12-00699-f005:**
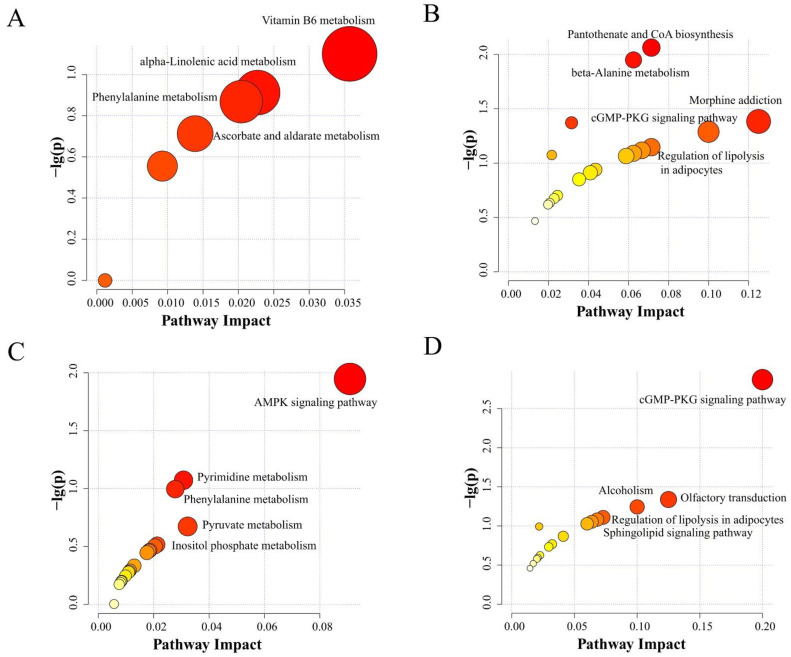
Milk KEGG pathway analysis of differential metabolites in the pairwise comparisons of LCaP compared to CON (**A**), MCaP compared to CON (**B**), HCaP compared to CON (**C**), and MCaP compared to HCaP (**D**). The x-axis is the pathway impact and the y-axis is the value of −log (*p*). The larger size indicates higher pathway enrichment and the darker color indicates higher pathway impact values. CON: control group, without calcium propionate addition; LCaP: low calcium propionate, the calcium propionate addition level was 200 g/d per cow; MCaP: medium calcium propionate, the calcium propionate addition level was 350 g/d per cow; HCaP: high calcium propionate, the calcium propionate addition level was 500 g/d per cow.

**Table 1 metabolites-12-00699-t001:** Effects of supplementing calcium propionate on the production performance of dairy cows in early lactation.

Items	Treatments				SEM	*p*-Value ^1^			
	CON	LCaP	MCaP	HCaP		L	Q	Time	Treatment × Time
DMI, kg/d	21.71	20.98	22.87	21.20	0.34	0.81	0.02	<0.001	0.57
BW, kg	781	758	767	753	4.79	0.10	0.63	0.46	0.99
ECM ^2^, kg/d	46.15	48.04	50.54	44.87	0.73	0.94	0.006	<0.001	0.61
4% FCM ^3^, kg/d	43.15	45.24	47.98	42.35	0.75	0.84	0.008	<0.001	0.54
Milk composition									
Fat, %	4.52	4.67	4.60	4.53	0.10	0.75	0.39	<0.001	0.64
Protein, %	3.35	3.29	3.15	3.15	0.04	0.007	0.83	<0.001	0.99
Lactose, %	5.13	5.01	5.00	4.99	0.02	0.003	0.09	<0.001	0.97
Fat/protein	1.36	1.43	1.47	1.41	0.03	0.44	0.32	0.56	0.54
SCC, 10^3^ cells/mL	201	135	72	151	14.61	0.09	0.03	0.65	0.84

CON: control group, without calcium propionate addition; LCaP: low calcium propionate, the calcium propionate addition level was 200 g/d per cow; MCaP: medium calcium propionate, the calcium propionate addition level was 350 g/d per cow; HCaP: high calcium propionate, the calcium propionate addition level was 500 g/d per cow; SEM: standard error of the mean; DMI: dry matter intake; BW: body weight; SCC: somatic cell count. ^1^ L = linear effect of treatment; Q = quadratic effect of treatment; Time = time effect; Treatment × Time = the interaction effect between treatment and time. ^2^ ECM (kg/d) = 0.327 × milk yield (kg/d) + 12.95 × fat yield (kg/d) + 7.20 × protein yield (kg/d) [22]; ^3^ 4% FCM (kg/d) = 0.4 × milk yield (kg/d) + 15 × milk fat (kg/d) [7].

**Table 2 metabolites-12-00699-t002:** Effects of supplementing calcium propionate on serum biochemical parameters related to NEB in dairy cows in early lactation.

Items	Treatments				SEM	*p*-Value ^1^			
	CON	LCaP	MCaP	HCaP		L	Q	Time	Treatment × Time
Glucose, mmol/L	2.83	2.94	3.15	3.09	0.01	0.02	0.49	<0.001	0.93
INS, mU/L	18.66	19.45	21.22	19.47	0.35	0.13	0.08	<0.001	0.70
Glucagon, pg/mL	72.58	73.12	73.65	73.29	0.30	0.29	0.50	<0.001	0.32
BHBA, mmol/L	0.88	0.83	0.81	0.87	0.01	0.32	0.003	<0.001	0.87
NEFA, umol/L	415	411	392	397	2.86	0.003	0.52	<0.001	0.73

CON: control group, without calcium propionate addition; LCaP: low calcium propionate, the calcium propionate addition level was 200 g/d per cow; MCaP: medium calcium propionate, the calcium propionate addition level was 350 g/d per cow; HCaP: high calcium propionate, the calcium propionate addition level was 500 g/d per cow; SEM: standard error of the mean; INS: insulin; BHBA: β-hydroxybutyric acid; NEFA: non-esterified fatty acid. ^1^ L = linear effect of treatment; Q = quadratic effect of treatment; Time = time effect, Treatment × Time = the interaction effect between treatment and time.

**Table 3 metabolites-12-00699-t003:** The significantly different milk metabolites of early-lactation dairy cows supplemented with different levels of calcium propionate.

Metabolites ^1^	*m*/*z*	RT (min)	VIP	*p*-Value	Log_2_FC ^2^
LCaP vs. CON					
4-Pyridoxic acid	182.04	136.29	2.86	0.008	−0.961
N-Acetyl-D-Glucosamine 6-Phosphate	300.03	186.54	2.06	0.046	−0.704
N-Carboxyethyl-gamma-aminobutyric acid	176.09	350.62	2.88	0.007	−0.515
sn-Glycerol 3-phosphoethanolamine	214.05	603.66	2.55	0.044	−0.358
D-Galactarate	191.02	434.33	2.11	0.047	−0.256
3-(3-Hydroxyphenyl) propanoic acid	165.05	114.32	2.42	0.026	−0.207
Traumatic Acid	227.12	65.98	2.44	0.033	0.285
Trp-Glu-Arg	490.24	225.62	2.80	0.011	3.952
MCaP vs. CON					
Genistein	271.06	33.10	1.80	0.040	−0.552
N-Carboxyethyl-gamma-aminobutyric acid	176.09	350.62	2.22	0.015	−0.451
Mevalonic acid	147.06	282.68	1.83	0.035	−0.437
1,3-Diaminopropane	116.11	278.44	1.80	0.049	−0.386
2,2-oxybis-Ethanol	107.06	104.19	1.94	0.037	−0.294
DL-Phenylalanine	207.11	287.83	2.13	0.032	−0.266
2-Methylglutaric acid	145.05	281.48	2.01	0.037	−0.240
Alpha-ketoisovaleric acid	115.04	137.54	2.00	0.031	−0.234
o-amino-Phenol	110.06	37.10	1.94	0.036	−0.226
Uridine 5-monophosphate	305.02	375.52	1.99	0.027	−0.218
D-(+)-Galactose	217.03	361.37	1.97	0.043	−0.098
Arachidonoylthio-PC	784.54	80.39	1.88	0.030	0.256
2-Deoxyadenosine 5-monophosphate	330.04	336.70	2.43	0.006	0.513
Leu-Phe-Arg	886.55	62.35	1.97	0.042	0.748
Adenosine	266.08	232.64	1.95	0.038	0.801
Uracil	113.03	221.32	1.79	0.048	1.060
Protoporphyrin IX	563.26	215.42	1.98	0.050	1.129
Sphingomyelin (d 18:1/18:0)	731.60	225.71	1.97	0.032	1.394
HCaP vs. CON					
Pro-Val	215.14	344.54	1.75	0.047	−0.872
D-Mannitol	163.06	244.69	1.97	0.020	−0.752
[2-(2-methoxyethoxy) ethoxy]-Acetic acid	179.08	38.85	2.24	0.017	−0.668
DL-lactate	89.02	224.54	1.88	0.035	−0.663
D-Glucuronate	193.03	436.08	1.92	0.030	−0.636
Deoxyguanosine	266.09	315.85	2.17	0.012	−0.481
4-Hydroxybenzoate	174.98	31.69	2.05	0.013	−0.460
Ile-Gly	189.12	336.57	2.43	0.005	−0.421
Uridine 5-monophosphate	305.02	375.52	2.55	0.002	−0.373
2-Methylglutaric acid	145.05	281.48	1.76	0.043	−0.226
Salicylic acid	137.02	92.52	1.91	0.029	−0.179
Berberine	355.11	437.77	1.81	0.044	−0.172
o-amino-Phenol	110.06	37.10	1.88	0.049	−0.169
Homocitrate	187.01	451.71	1.91	0.048	−0.161
Gentisaldehyde	138.02	179.25	1.80	0.043	−0.147
Barbituric acid	128.00	110.17	1.87	0.046	−0.120
Quercetin	301.03	690.58	1.90	0.039	−0.118
D-(+)-Galactose	217.03	361.37	1.96	0.028	−0.112
Kynurenic acid	188.03	283.76	1.90	0.035	0.275
NG, NG-dimethyl-L-arginine	203.15	536.72	2.14	0.018	0.302
3-Aminopyridine	95.06	495.16	2.11	0.013	0.330
Protoporphyrin IX	563.26	215.42	2.27	0.009	1.382
MCaP vs. HCaP					
1-Octadecyl-2-acetyl-sn-glycero-3-phosphocholine	552.39	209.58	2.49	0.008	−1.228
1-Hexadecyl-sn-glycero-3-phosphocholine	482.34	251.30	2.04	0.035	−0.828
1-Hexadecanoyl-sn-glycero-3-phosphoethanolamine	452.27	222.12	1.82	0.047	−0.768
Adenosine	266.08	232.64	2.30	0.022	−0.711
C16 PAF	524.36	237.89	2.24	0.020	−0.711
[2-(2-methoxyethoxy) ethoxy]-Acetic acid	179.08	38.85	2.28	0.011	−0.586
Guanosine 5-monophosphate	382.05	477.10	2.05	0.014	−0.444
Lanosterol	427.37	61.29	2.35	0.017	−0.368
DL-lactate	89.02	224.54	1.93	0.032	−0.320
Ile-Gly	189.12	336.57	2.31	0.017	−0.293
Quinate	191.05	585.58	1.98	0.049	−0.122
N-Formylmethionine	176.04	248.86	1.77	0.040	0.192
NG, NG-dimethyl-L-arginine	203.15	536.72	2.10	0.039	0.269
Kynurenic acid	188.03	283.76	2.30	0.027	0.294
N6-(1-Iminoethyl)-L-lysine	205.15	592.90	2.06	0.043	0.342
Raffinose	522.18	482.26	2.33	0.012	0.350
D-Quinovose	201.02	608.35	2.11	0.034	0.403
trans-cinnamate	149.05	329.36	2.47	0.009	0.463
N-(4-Hydroxyphenyl) propanamide	232.02	35.08	2.06	0.028	0.516
Leu-Gly	188.12	196.72	2.24	0.024	0.597
1,5-Diaminonaphthalene	158.09	199.10	2.10	0.035	0.616

*m*/*z*: mass to charge ratio; RT: retention time; VIP: variable importance in the projection; FC: fold change; CON: control group, without calcium propionate addition; LCaP: low calcium propionate, the calcium propionate addition level was 200 g/d per cow; MCaP: medium calcium propionate, the calcium propionate addition level was 350 g/d per cow; HCaP: high calcium propionate, the calcium propionate addition level was 500 g/d per cow. ^1^ The differential metabolites were screened with the thresholds of *p* < 0.05 and VIP > 1.00. ^2^ A value of Log_2_FC > 0 indicated the metabolite was upregulated, while Log_2_FC < 0 indicated the metabolite was downregulated.

**Table 4 metabolites-12-00699-t004:** The ingredients and nutrient composition of the basal diet.

Items	Value
Ingredients	
Sprouting corn bran, g/kg DM	21.88
Stem-flaked corn, g/kg DM	35.77
Cottonseed, g/kg DM	22.75
Megalac ^1^, g/kg DM	4.95
Fat power, g/kg DM	11.43
Pelleted beet pulp, g/kg DM	13.05
Wet brewer grains, g/kg DM	37.32
Alfalfa, g/kg DM	99
Oat hay, g/kg DM	21.63
Concentrate ^2^, g/kg DM	419
Corn silage, g/kg DM	313
Nutrient composition, % (*w*/*w*) of DM	
CP	17.7
NE_L_ ^3^, MJ/kg	7.20
aNDF ^4^	28
ADF	15.9
EE	4.2
Ca	0.85
P	0.42

^1^ Megalac is a rumen-protected fat supplementation produced by Volac Wilmar Feed Ingredients, Ltd. (Hertfordshire, UK). ^2^ The concentrate for postpartum dairy cows was manufactured by Beijing Sanyuan Seed Technology Co., Ltd. (Beijing, China). The concentrate nutrient contained: DM, 88.50%; CP, 23.91%; aNDF, 13.20%; ADF, 7.40%; Ash, 13.1%; Ca, 1.41%; P, 0.58%; K, 1.20%; Mg, 0.58%; Na, 0.99%; Cu, 46.25 mg/kg; Fe, 80.30 mg/kg; Zn, 136.76 mg/kg [14]. ^3^ The net energy for lactation (NE_L_) was a calculated value according to NRC [7]. ^4^ Neutral detergent fiber was assayed with heat-stable amylase and is expressed inclusive of residual ash.

## Data Availability

The data presented in this study are available in Appendix A.

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
