# Peer review of "Alterations in the Milk Metabolome of Dairy Cows Supplemented with Different Levels of Calcium Propionate in Early Lactation"

_metabolites, 2022, doi:10.3390/metabo12080699_

Round 1
Reviewer 1 Report
Dear Authors
1. The paper is generally good.
2. On line 16: The term LC-MS needs to be clarified.
3. On line 71: It's better if it's written like this: Dairy cow diseases.
4. On line 114: In Table 1, are the values given in g/kg of DM as the composition of the ration? Here it is better if the unit g/kg DM is placed below the value.
5. On line 140: the number in brackets (4:3:3) not clear, that needs to be clarified.
6. On line 144: the authors calculated the following model for energy-corrected milk yield (ECM) (kg/d): ECM = 0.327 × milk yield (kg/d) + 12.95 × fat yield (kg/d) + 7.20 × protein yield (kg/d). Usually the following model was used: Energy corrected milk (ECM; kg) =kg milk production x (0.383x milk fat%+0.242 x milk protein%+0.7832) / 3.1138. The question is, is there a reason for this?
7. On line 230: The statistical model needs to be written as follows:
Yijk = µ + αi + βj + γk + αi × βj + εijk
8. On line 281; Figure 1: The authors did not mention this figure in the text and did not explain its content.
9. The figures in the text are not placed in the correct order. These must be placed in the correct order.
10. There are contradictions in the explanation of the results, namely: in lines 416 to 418 the authors wrote that treatment with calcium propionate could lead to more glucose being produced in the liver for lactose synthesis and in line 424 the feeding of calcium propionate reduces the milk protein and milk lactose concentrations in the milk produced. Is there an explanation?
11. On line 424: the authors wrote that feeding calcium propionate reduced milk protein and milk lactose concentrations in the dairy cows, which may be related to increased milk production. This explanation is not convincing. We know that lactose is constant in the milk because it plays a very important role in udder osmotic pressure and is rarely influenced by feed. It is also worth knowing that an increase in milk yield affects the fat and protein content of the milk. However, the results show something completely different, since the increase in milk yield increased the fat content of the milk and only the protein content decreased. I suspect a feeding problem. If the urea in the milk is determined, we can determine what is the problem with the protein in the milk and in the ration.
12. On line 555: There are two repeated words. These are: was conducted.
13. There are a few notes in the reference list:
- The reference 16 are two references and not just one. This means that the following series has to be changed and also in the text as the number has to be changed.
- Reference 20 does not have a page number at the end of the studies.
- The reference 23 was published in 2021 and not 2020. This is an online publication.
I hope that these advices will be taken into account.
I wish you success
Reviewer 2 Report
A very interesting topic. The manuscript is of high quality, as is the methodology and discussion of the results.
I only have a few suggestions for corrections:
p. 1, line 20, 21: diets instead of dietary
p. 2, line 43: increase instead of increased; add a comma after liver; line 47: insert a before positive; line 71: insert a comma after lactation
p. 3, line 97: insert ";" after replicates; line 106: begin a new sentence after ...18:00). The ingredients...
p. 4, line 133: insert "of" before residual
p. 5 and 6, lines 173, 177, 183, 191: check °C for consistency
p. 6, line 205: bewere? Meaning not clear.
Figures 1, 2 and 3 should be checked and figures 2 and 3 should be swopped. The figure should be included after it has been mentioned in the text.
p. 10, line 303: insert "to fact" after related to; line 305: include reference to Figure 1.
p. 11, line 350: include "Figure 2" here.
p. 12, line 367: should be Figure 3?
p. 16, line 401: insert "the observation" before that...; line 423: inhabiting instead of inhabited; line 428: Supplementing instead of Supplementation; line 429: insert "a" before calcium; line 430: ...immunity, which...
p. 17, line 473: acids; line 475: in vitro; line 476: ...mammalian cells [43]. The...
p. 18, line 493: produced instead of production
Reviewer 3 Report
The study examined the effects of feeding 3 levels of calcium on the production performance of dairy cows and the metabolite profiles of their milk. Despite the volume of data present in this manuscript, the issues in data analysis and interpretation prevent its acceptance in its current form.
1. The samples were collected at 5 different time points (day 7, 14, 21, 28, 35 postpartum), but the values of production, serum, and milk parameters in table 1, 2 (also SIMCA-P) appear to be the least square means from SAS (mentioned in 2.6, but not stated in respective tables). This practice is troublesome in both data presentation (one value for 5 time points) and biology. Day 7-35 is in the middle of transition period, in which dairy cows went through dramatic changes in all these parameters. One value per treatment in these tables does not reflect the real status of production, serum biochemicals, and milk metabolites. The P values of time effect in the tables do not provide a good idea on the changes of individual parameters during this period. In addition, the time-dependent changes in the metabolome are not shown in the models (Figure 1).
2. Instead of OPLS-DA models on individual pairs of two treatments, a model of 4 treatment groups should be produced (such as PCA or PLS-DA model) and interpreted for a more comprehensive view on the metabolic effects of calcium supplementation as well as the time-dependent effects in the transition period. The separation in OPLS-DA is highly biased and could be artificial since almost any two sample groups can be separated by OPLS-DA for minor differences.
3. The LC-MS analysis has no coverage on the lipids, which are the staple component in milk. In addition, the excel file in the supplementary info lists the metabolite markers as "MS2 matched". Not sure how their identifies were matched and confirmed since this is not reflected in 2.5.2. All these cast the concerns on the validity of pathway analysis and discussion.
Round 2
Reviewer 3 Report
I appreciate the authors' efforts in addressing my concerns. New Figure 1 on 5 time points and new Figure 2 on 4 treatment groups are far better than the previous presentation. There is still space for improvement in metabolite analysis, including the lack of lipid species and structural confirmation, but I can accept its current form for practical reasons (service cost). Please conduct a careful proofreading.